# Training providers to implement heart failure shared medical appointments: A qualitative evaluation

Rene Hearns[1,‡]*, Sherry L. Ball[1,‡], Tai-Lyn Wilkerson[1], Julie Gee[1], Sharon LaForest[1], Kimberley Schaub[1], Tracey Taveira[2,3], Wen-Chih Wu[2,4]

1 VA Northeast Ohio Healthcare System, Medicine Service, Cleveland, OH, United States of America, 2 VA Providence Health Care System, Medicine Service, Providence, RI, United States of America, 3 The University of Rhode Island, College of Pharmacy, Kingston, RI, United States of America, 4 Department of Medicine, Alpert Medical School & Department of Epidemiology School of Public Health, Brown University, Providence, RI, United States of America

☉ These authors contributed equally to this work.
‡ RH and SB are Joint Senior Authors
* rene.hearns@va.gov

**Data Availability Statement:** The United States Department of Veterans Affairs requires that a Limited Dataset (LDS) will be created and shared pursuant to a Data Use Agreement (DUA)

## Abstract

Shared medical appointments (SMA) help patients learn skills to self-manage chronic medical conditions. While this model of care delivery is thought to improve access to care with an efficient use of healthcare providers' time, many healthcare teams struggle to implement this healthcare model. Guidance and training resources on the implementation of SMAs is expected to improve adoption, implementation and sustainability of SMAs. Our evaluation team collaborated with the HF SMA trainer to complete a developmental formative evaluation of a two-day training program with a goal of adapting the training program and to better suit the needs and resources of healthcare teams interested in implementing HF-SMAs. Our evaluation team interviewed members of healthcare teams participating during each stage of training: pre-training, post-training, and post-implementation. The evaluation team also observed training sessions and reviewed minutes from debrief and training team meetings. Qualitative data collected from interviews, observations and document reviews were analyzed using matrix analysis with a focus on identifying potential adaptations to improve the HF-SMA training program. Data summaries were presented by the evaluation team to the training team for consideration. Training program participants found the training comprehensive, useful, and helpful; they highlighted how the trainers were experienced SMA providers who shared lessons learned. While participants found the training to be useful, opportunities for improvement, success of the virtual format and identified six adaptations: 1) the two-day training was reduced to short online modules, 2) curriculum was adapted to fit local site's needs, 3) added periodic one-on-one coaching, 4) shifted training to focus on skills and knowledge needed for each team member requirements, 5) adapted curriculum provides for more team building during pre-meeting assignments, and 6) training had more information included. We offer/present an improved model for a HF-SMA training program. Future studies, potentially using comparative designs to measure success and sustainability are needed.

appropriately limiting use of the dataset and prohibiting the recipient from identifying or re-identifying (or taking steps to identify or re-identify) any individual whose data are included in the dataset. Contact the VA Cleveland Information System Security Officer via email at: vhacleisso@va.gov.

**Funding:** VA Health Services Research & Development grant # IRP 20-003 entitled: Implementation Trial Evaluating On-site In-person Versus Remote Video- Assisted Facilitation to Train Providers on the implementation of Shared Medical Appointments (SMAs) in Heart Failure (HF) or HF-SMAs. Principal Investigator: WCW; Co-Investigator: SB. The funder had no role in study development, design, data collection, analysis, decision to publish, nor manuscript development.

**Competing interests:** The authors have declared that no competing interests exist.

## Introduction

Although diagnostic and therapeutic advances for heart failure (HF) continue to grow, its prevalence continues to increase. Approximately 6.2 million people in the United States are living with heart failure [1] and approximately one person will die from heart failure every 33 seconds [2]. One key to decreasing heart failure-related mortality lies in the patient's ability to self-manage their condition. When patients with HF learn about and adopt a healthy lifestyle that for example, includes smoking cessation, weight management, proper nutrition, and exercise, disease progression is slowed [3].

In contrast to the conventional one-to-one clinic structure, shared medical appointments (SMAs) teach patients self-management skills and provide enhanced muti-disciplinary care in a group setting. In 2010, the Veterans Health Administration encouraged SMAs as means to improve clinic efficiency and quality. By accommodating multiple patients into one appointment, SMAs can reduce wait times and improve clinic utilization [4]. HF-SMAs support improved HF management through patient education about medication and improved access through scheduling flexibility and more appropriate health care utilization [5]. For patients recently discharged, HF-SMA attendance resulted in improved quality of life measured by EQ-5D (co-primary outcome) compared to usual care [6].

HF SMA participants, including patients and providers, shared that they found these SMAs improved clinical outcomes, patient-caregiver-provider interactions, peer support, and self-care skills [7]. Providers and patients reported high satisfaction as SMA leaders [8] and participants [9], respectively. Therefore, HF-SMAs benefit patients by improving their quality-of-life, social support, and satisfaction with their health care providers. Providers benefit by improving work satisfaction and clinic efficiency.

Despite the known advantages of HF-SMAs and the growing prevalence of HF, only a small portion of HF patients are offered care in an SMA due to limited implementation of HF-SMA clinics [10,11]. Training providers could help increase the number of SMAs offered but few provider-focused HF-SMA training programs exist within the VHA. Training barriers include scheduling providers and training time, the organization of the training, setting goals, coaching, provider roles, site team's cohesiveness, and missing pieces within the training [12]. Existing teams running successful SMAs have the knowledge and expertise to initiate and sustain SMAs, but little guidance is available regarding how to best disseminate and spread these practices.

Our study aimed to identify adaptations to an existing training program for diabetes SMAs [13], as a case study and a model for HF teams, through the examination of trainee and trainer experiences of the adapted training program and to inform future implementation [14]. We sought to observe and refine these SMA leaders' approach to training guided by the SMA sites. We used the Quality Enhancement Research Initiative (QUERI) implementation roadmap to guide data collection in our qualitative study.

## Materials and methods

### Study design

We completed a developmental formative evaluation of a training program to teach clinical teams to setup and implement heart failure shared medical appointments (HF-SMA). The training program's curriculum consisted of four parts: 1) didactic presentations, 2) needs assessment/question and answer session, 3) virtual observation of an actual HF-SMA, and 4) simulated trainee-led SMA using standardized patients.

**Table 1. Data categories & subcategories.**

| Category | Operational Definition | Subcategories | |
|---|---|---|---|
| Physical Space & Technology | Physical location and technology available and/or used for the training | Addl. Assistance /Tech Clinic Location Computer Troubles Difficulties Equipment | Room setup Switching Technology Technology Technology Connection |
| Preparation | Any training materials shared with trainees and /or used by trainers that was created and available prior to the training, includes previous training as trainers with the presentation technology, have a prepared presentation, and arranging to have appropriate staff involved in the training. | Administration (IRB, schedule) Clinic Flow Meeting Preparedness | Preparation Room Setup SMA planning Technology Preparation |
| Group Dynamics | Anything that could facilitate or inhibit the ability of the trainers and trainees to interact with or between each other | Benefits Engagement Introductions Managing Group Patient Rapport | Questions Role of Team Member Sharing Skills SMA Planning Team View |
| SMA Appointment | SMA Appointment refers to training elements that are specific to the training content and its structure | Administrative Ideas Clinic Flow/Timing Confusion/Difficulties Education Med Rec/Adherence Packets for Patients Patient History/Status Patient Population Program Importance | Questions for Patient Scheduling SMA Documentation SMA follow-up SMA Referral Teaching Tools/ Education Training Presentation/ Platform |
| Training Specific | Content and/or structure of the training | Communication Debrief Difficulties Feedback from Trainees Mentoring Mock Huddle Mock SMA Observation Methods Program Adaptation | Program Importance Reflection Timing Trainee Questions/ Confusion Training Flow Training Presentation/ Platform Training Questions |
| Future ideas | Any identified adaptations to improve future rollout of training program | Adaptations Future Ideas | Future SMA SMA Goals |

The study team was a collaboration between, the training team (providers) and the evaluation team (researchers). The training team was composed of experienced HF-SMA leaders from two sites who worked together to develop a two-day training program. The training team consisted of one MD, one NP, two PharmDs, and one psychology PhD (1 male and 3 females).

The evaluation team, not part of the training team, was led by a qualitative scientist (PhD) and four experienced qualitative researchers, (all female) with over 20 years of prior experience with evaluation/SMAs, who collected data through interviews, observations, and/or performed document review. The training team led all training sessions. The two teams met regularly to discuss recruitment, data collection, interpretation, and potential adaptations to the training; teams also conducted debriefing sessions after each training activity.

The QUERI implementation roadmap provided a framework for data collection [15]. The pre-implementation phase data was collected through interviews with trainees. The post-training phase data was collected through interviews with the trainees, document review of debriefs and team meetings, and observations of training sessions and implemented HF-SMAs. Sustainment data collection included interviews with trainees. The data from pre-training interviews were organized into a categories and subcategories based on the (Table 1). These

Table 2. Initial HF-SMA training curriculum.

| Initial HF-SMA Training Curriculum | |
| --- | --- |
| Day 1 | Day 2 |
| 5 hour didactic presentation:<br> 1. Overview of SMAs: identifying benefits, leadership support; patient population; goals and guidelines, and space requirements.<br> 2. Getting Started: establishing a clinic, identifying billing codes, teaching techniques, etc.<br> 3. Key components of the SMA: pre-appointment huddle, group discussion, prescriber breakout, and post-visit summary.<br> 4. Implementation Tools: included 20 Supplemental Resources | Observed an Actual HF-SMA<br> 1. HF-SMA pre-appointment huddle<br> 2. HF-SMA: group discussion and prescriber breakout<br> 3. HF-SMA post-appointment huddle |
| 1. Question and Answer Session<br> 2. Discussion of Day 2 training | 1. Simulated HF-SMA with standardized patients.<br> 2. Question and Answer Session |

identified categories and subcategories were used to create a matrix template [16] with a focus on identifying potential adaptations [17] to the training program. Research involved human participants has been performed in accordance with the Declaration of Helsinki and approved by the VA Northeast Ohio Healthcare System's IRB project CY19-028 on January 10, 2020 and required written consent.

**Training curriculum.** We used an experience-based interactive virtual learning curriculum. The training program incorporated didactic sessions, real-time virtual observation of an ongoing HF-SMA clinic run by the program's training team, and use of simulated patients during trainee practice sessions. The training team developed the training curriculum based on best practices for creating and sustaining a HF-SMA [5,7] in alignment with VA MISSION Act to improve access to specialty care [18], with the goal to train future clinical teams to implement HF-SMAs. The initial training curriculum is summarized in Table 2.

**Recruitment.** The site recruitment period was May 8, 2019 through June 15, 2022. A general email inviting sites to participate in the training were distributed through a National VA HF network listserv on May 8, 2019. The email included information regarding the two-day training that would enable the site to create a HF-SMA for their site. Four sites volunteered to participate by contacting the study investigators. There was no prior relationship with the sites. Two sites were selected based on the study team's access to the site champion. A site champion is a cardiac medical provider who will champion with leadership and guide the SMA team through the process. The study PI approached the trainee sites' champions to gather information regarding each site's leadership support, expertise, and resources available for the implementation of HF-SMA. Both sites were interested in improving their access and reducing rehospitalization rates through specialty care utilization of the HF-SMA. The evaluation team contacted 14 adult trainees (providers and staff) from the two selected sites, obtained written informed consent from 10 individuals, and scheduled interviews with eight.

**Interviews.** Guided by principles from the field of adaptation [19], we identified and incorporated the resources and needs of trainees and tailored the training to meet the local needs of the patient population the trainees serve. The interview guides were developed by the evaluation team based on the Promoting Action on Research Implementation in Health Services (PARiHS) [20] framework (S1–S3 Appendices). The interviews of the recruited sites were between 2/2/2020 through 11/23/2020. The participants provided written consent and the interviews were one-hour and completed pre-training (2/2/2020-2/5/2020), post-training (2/19/2020-3/5/2020), and post-HF-SMA implementation (11/20/2020-11/23/2020). All interviews were audio recorded and transcribed. A codebook was developed based on an inductive analysis of the pre-implementation interviews.

**Observations.** Observations were of 1) two-day training and 2) trainee site's implemented HF-SMA clinics. The two-day training program observations were completed by two evaluators (one in-person and one via virtual connections) to understand the experience of both perspectives. The evaluation team used the training observation guide (S2 Appendix) to organize their note taking. After each observation, the two evaluators met to reflect and discuss the data collected. The trainee sites' implemented HF-SMA clinics were observed via virtual connections. The trainee sites' HF-SMAs were observed for fidelity to the training.

**Document review.** An evaluation team member reviewed meeting minutes, asynchronous meeting follow-up communications (emails) and placed the review notes into the matrix for analysis. Observation data were organized chronologically noting time, observer, person being identified in observation, training progression, and observer notes.

**Qualitative analysis.** The evaluation team analyzed the data using matrix analysis in MS Excel to identify potential adaptations to the HF-SMA training program. Each evaluator coded data and the coding was iteratively discussed in evaluation team meetings to identify recurring issues and potential adaptations. Adaptations were discussed until consensus was reached on what was to be adapted and prioritized. Analysts shared identified adaptations with the training team at monthly team meetings and incorporated the team approved adaptations into the training program. Finally, the evaluation team reviewed the data for thematic analysis that lead to the themes as presented in the results.

Primary data (observations, emails, and meeting minutes) were placed into the matrix. The matrix's column headers were: Category; Subcategory. The list of category and subcategory codes was discussed for consensus on operational definitions of each domain. Three evaluation team members independently completed the same matrix and compared their results. After each data collection, the evaluation team iteratively reviewed the data, what worked best and potential adaptations, reached consensus, collected recommended adaptations, and shared proposed adaptations to the training team. The data were reviewed for themes and discussed with the training team.

## Results

Table 3 provides the project's timeline, data quantity, and type collected. The initial iteration of the training included two sites, where Site-1 was an in-person training and Site-2, a virtual training. However, Site-1 withdrew due to the travel restrictions and competing demands associated with the COVID-19 pandemic, which left only Site-2 for synchronous virtual training according to the initial curriculum (Table 2). Based on data collected from the initial iteration, we identified travel and time barriers to attend an in-person training in addition to the event sequence of the curriculum. These initial barriers were overcome through revising the curriculum into a remote format of both synchronous and asynchronous sessions to increase time flexibility for the provider and improve the training efficiency through event re-arrangement. In the second iteration, a remote site assessment occurred first followed by an online asynchronous training and supplemented by weekly half-hour trainer/trainee synchronous meetings for discussion and to answer questions. Three additional sites were recruited, and the adapted training structure was used.

The participants found the training comprehensive, useful, and helpful; they highlighted how the trainers were experienced SMA providers who shared lessons learned. While participants found the training to be useful, our analysis revealed further opportunities for improvement. The training structure and content was adapted from the Site-2's feedback and the adapted version was used through the fifth site. All the sites felt that the training would facilitate implementation of HF-SMAs and lead to improvements in access and patient outcomes.

**Table 3. Project timeline and data summary.**

| Item | Timeframe | # | Data Collected |
|---|---|---|---|
| Training team preparation | 11/2019 thru 01/2020 | | |
| Study Team meetings | Monthly– 1/2020-7/2022 | 21 | Meeting Notes |
| Pre-training Interviews | 01/2020 thru 2/2020 | 5 | Interview transcripts |
| Site-1 two-day in-person training | 01/2020 | | Postponed due to travel delays/ then canceled due to COVID-19 |
| Site-2 two-day virtual training (didactic, needs assessment, HF-SMA clinic observation, mock SMA) | 02/2020 | 2 | Observations |
| Evaluation Team debriefs | 02/2020 | 2 | Notes |
| Study Team debrief | 02/2020 | 1 | Meeting Notes |
| Post-training Interviews | 02/2020 thru 03/2020 | 5 | Interviews |
| COVID 19 | | | |
| Training team one-to-one coaching | 03/2020 thru 03/2021 | | No data collected |
| Recruited sites 3–5 | 08/2021 thru 05/2022 | | |
| Implemented revised training; Sites 3–5 | 08/2021 thru 06/2022 | | Notes |
| Site-2 implemented HF-SMA | 10/2020 thru 06/2022 | 4 | Observations |
| Post-implementation Interviews | 02/2022 thru 04/2022 | 3 | Interviews |
| Site Needs Assessments; Sites 3–5 | 12/2021 thru 06/2022 | 3 | Notes |
| Site-5 began adapted training | 06/2022 thru 08/2022 | | |
| *Project's funding ended* | 06/2022 | | |
| Site-5 implemented HF-SMA | 09/2022 | | |

### Site-1

This site was part of the original two and was supposed to have the in-person training. However, due to administrative delays, it was postponed until after the virtual training. Subsequently it was cancelled due to COVID-19's travel restrictions.

### Site-2

The second training site training provided the most adaptations. The revised training program was used to assess three additional sites (Sites 3–5) for infrastructure preparedness through the needs assessment. Findings from Site-2 are expanded upon in the Qualitative Summary section. Site-2 implemented and maintained a HF-SMA clinic throughout the project. Their HF-SMA clinic provided recently hospitalized patients quick access to clinical follow-up and self-management education. Efforts to overcome barriers of sustaining their HF-SMA clinic included staffing changes, patient recruitment challenges, and facility leadership changing priorities.

### Site-3 and Site-4

After the initial site assessment and discussion with the sites, both sites needed to develop a HF clinical practice prior to implementation of a HF-SMA. It was agreed that an established HF clinical practice serves as a referral source and clinical support for the HF-SMA and was deemed important by the training team for successful implementation and sustainment.

### Site-5

A fifth site was assessed and completed the training, but post-training interview was not completed due to project ended (June 2022). This site was able to hold their first HF-SMA in mid-September 2022. When comparing the original training (trainee's time to implementation) to

the reorganized training, the reorganized training reduced the time by five [5] months for Site-5 to hold their HF-SMA clinic. Additionally, under the revised training, the trainees were able to add documents from their site to further expand the available information available for future trainees.

## Qualitative summary

**Assessment and adaptations.**    *Theme*: *Trainees' general impressions of training*. Participants emphasized the value of having trainers who were experienced providers from multiple disciplines and were able to share valuable lessons was "really nice." Seeing how the trainers ran an actual SMA session helped solidify the SMA workflow and helped participants think through what type of issues may arise. Finally, the progressive nature of the training facilitated learning and provided independence for the site.

> "I thought most helpful and valuable was hearing about kind of what they've learned through the years and what they've improved upon. Kind of knowing somethings to avoid going into it and help ours be successful from the start." Post-training S010.

> "we received a nice overview of the various aspects of shared medical appointment from the different specialty providers." Post-training S007.

> "it was really helpful to see the sample or to watch how a day with the SMA [actual clinic], how the flow was" Post-training S009.

> "really nice to see some of the potential issues that could come up and during the actual patient appointments there was someone that got admitted and so we got to debrief that so I think going over the things that can go wrong for me is super useful because that's what going gum up the works when we're trying to implement something like this." Post-training S008.

> "I think the way it [training] address[es] the learning in a sequential manner and that each part of that training was more and more trying to push us to independently think about it [the HF-SMA process]. . .. helping us, facilitating our learning as well as like us coming up with our own ideas, I think that was done very well." Post-training S006.

*Theme*: *Difficulty in scheduling two-day training*. Both the training team and trainees noted challenges to scheduling and attending all sessions due to the providers clinical responsibilities.

> "having two days . . . set aside for this training is hard, because that means that the staff is unavailable for patient care" Pre-training S006.

> "split that nuts and bolts to maybe shorter sessions leading up to that actual SMA" Post-training S006.

> *Training team indicated one hurdle was the 2-day training format and decided that the training should provide the information over a longer period of time. Furthermore, the discussion included possibly recording videos for asynchronous learning, expanding the pre-work to reduce trainee provider burden, and provide short videos (no longer than 5 minutes) that provide the essential information from the didactic presentation with detailed information placed in the training manual.* Team Training Debrief minutes

Adaptation: The concentrated trainee time commitment was restructured from two full days to 14 short videos (5 minutes or less) that could be viewed at an asynchronous self-defined pace via MS Teams that minimally interfered with a clinician's schedule. To enhance the online training, 30 minutes or less weekly meetings were added for the trainers to answer questions and/or provide further direction.

*Theme*: *Trainers found difficulty with the organization of the training*. The original training curriculum had the following order: didactic presentation, site assessment, view HF-SMA clinic, and execute mock HF-SMA. The trainers considered the completion of a needs assessment should be done first before the training; therefore, the training can be adjusted to meet local facility needs.

*Trainers discussed changing sequence of training: Begin with recorded video tape of an actual SMA–trainees observe asynchronously–recorded, Give assignment in a workbook, etc.* Team Training Debrief minutes.

*Pompt trainees to complete needs assessment prior to training and use assessment to inform training; direct trainees to videos prioritized according to site's needs*. Team Training Debrief minutes.

Adaptation: Realizing the importance of making the didactic presentation relevant and tailored to the needs of the trainees, the presentation and needs assessment order was reversed.

*Theme*: *Local site's difficulty setting SMA goals*. The trainees indicated that they would like more assistance with goal setting. The trainers noted the importance of allowing the teams to work together early on to set goals for their own SMA clinic.

"I think maybe helping the team flush out like what exactly are the goals of the SMA for that particular medical center and maybe a little bit guidance on how to structure the SMA. . ." Post-training S007.

*Training team felt that they needed to review and revise curriculum's pace and goals*. Team Training Debrief minutes.

Adaptation: The new training structure would allow teams to independently identify the appropriate clinic goals including the target population and identify or train the appropriate clinical team members.

*Theme*: *Trainees felt that they needed more coaching*. Trainees asked for more coaching to go through the entire process. They wanted more interaction throughout their start-up phase.

"training is really more of a process, and it shouldn't be seen as like a one-time implementation with a couple of phone calls. It should really be more of a continuous type of interaction." Post-training S007

"I think you guys [trainers] are great to [volunteer to] meet with the team like every two to three weeks or whatever follow up is as we roll out the SMA on our side to try to keep us on track" Post-training S006.

*Add individual coaching sessions by listening to the trainees and providing the training team' insights as to how each training site may address the issue during the training team/trainees meetings*. Team Training Debrief minutes.

Adaptation: Adding one-on-one coaching during the weekly trainer/trainees meetings to enhance the self-paced smaller learning components from the prior didactic presentation. This adaptation is expected to improve team cohesiveness and reduce the training time burden.

*Theme*: *Confusion with HF-SMA team member roles*. The trainees had not selected a moderator/lead for the HF-SMA and indicated confusion regarding the team's roles.

"I think some direction as far as the role of the moderator would be helpful. . .I think the role of the moderator is really important and I think that the type of background that the moderator should have should be kind of clearly laid out" Post-training S007.

*The notes indicated that the skillset of the roles, especially the roles of the provider and moderator, needed to be highlighted with an expanded explanation of each role's skills and importance. They believed that the moderator was the least defined role and that the trainees need to understand that lecturing needs to be replaced with the use of probes like 'Tell me' or 'Show me.'* Team Training Debrief minutes.

Adaptation: In the training manual section "Identify HF-SMA Core Team Members" now provides more detail regarding each provider's role. The appendix of the manual now includes sections detailing the moderator's role and group dynamics. The workbook was appended to include a section to guide trainees on how to select providers for each role (facilitator/moderator, provider, medicine reconciler, etc.).

*Theme*: *Lack of team cohesiveness*. The training team felt that the trainees needed more time to develop their team. The trainees disclosed the difference between working together and being a team.

"[Training was] trying to push us [as a group] to independently think about it [their future SMA]. . .. helping us, facilitating our learning as well as like us coming up with our own ideas, I think that was done very well." Post-training S006.

"I think getting us all together as a group to do the training helped just kind of establish a team we all, even though we work together a lot of the times we are doing our own thing so getting us all together as a group kind of helped facilitate teamwork and helped us to approach this together, starting this together." Post-training S009.

*The training team felt that the trainees needed to develop their team by working on assignments together and then discussing the results with the training team. It was determined that having a MS Teams group for the asynchronous items and easier access to the materials and a workbook would assist the trainees with building their team*. Team Training Debrief minutes.

Adaptation: Trainers will direct teams to complete and engage with the asynchronous training first, then huddle to organize their questions and participate in the trainer/trainee sessions where trainees would get clarity on any questions as a team.

*Theme*: *Missing pieces*. Post training interviews highlighted a need to include specific practical resources to facilitate successful HF-SMA implementation HF-SMA and reduce the time between training and implementation.

## Note templates

The trainees requested copies of clinical note templates to assist with their clinic documentation.

Adaptation: Both Cleveland and Providence's clinical note templates were sent to the trainees and added to the manual as appendices.

## Workbook

The training team discussed the need of pre-assignments for various topics to assist the trainees with developing a better understanding of the topic.

*Reflecting on the notes taken during the training, each training team member indicated that more work needed to be completed in advance by the trainees. Not only will the prework assist with their team development, but it will also provide them with a topical foundation for which to ask the training team meaningful questions.* Team Training Debrief minutes.

Adaptation: A workbook was created with five group exercises for HF-SMA Team, Clinic Goals, Education Information, Pre-Clinic Huddle, and Mock HF-SMA.

## Goals

The training team and trainees felt that the SMA's goals needed further prework and more clarification within the manual as to how to establish the goals that are appropriate for each new site.

"I think maybe helping the team flush out like what exactly are the goals of the SMA for that particular medical center and maybe a little bit guidance on how to structure the SMA and the reason I say that is you know there are different approaches to kind of the type of patients are referred" Post-training S007

Adaptation: The training team included examples of possible goals, metrics spreadsheets, and how to attain the goals in the manual; along with a section in the workbook that provides guidance on goal development.

## Patient identification

The trainees discussed difficulties with identifying which patients should be in the HF-SMAs and the associated process.

"I guess the main thing is trying, figuring out, how you get all of these people [patients] in a group" Post-training S010.

"A part of it [the training] could be a little bit more rigorous support in terms of identifying the right patient that can [be] help[ed by attending an SMA]." Post-training S004.

"once, the LPN identifies a patient, someone needs to read through the chart to see, OK, do they really meet the criteria to be in a group? Are they blind? Are they, do they have dementia? Are they nursing home patients? Are they hard of hearing because those kind of patients probably not appropriate for this group" Post-training S009.

Adaptation: The manual was adapted to include more guidance on population management, patient selection, colleague support/referrals, reviewing patients, and utilizing the HF Dashboard. Multiple videos were created to address the important topics.

### Additional resources

As the adaptations were incorporated into the training program, the training team identified the need for additional resources that would enhance the training program.

*The training team and medical experts provided information on what resources would help a team when starting an HF-SMA to reduce the start-up burden. For example, templates for presenting the education, administrative information, patient handouts from heart failure societies.* Various Emails and Observation notes.

- Education Presentation Templates:

   The training team' presentations were adjusted to become templates for trainee site to start the HF-SMA and adjust as necessary. These covered four topics of HF-Symptoms, Nutrition, Medication, and Summary (teach back). The presentations were adapted for any site to add site specific information.

- Educational Handouts:

   National association handouts for patients were added, as well as the HF Education Handbook created as a workbook for the patients.

- Administrative Information:

   Additional information was added to the manual regarding creating a clinic, scheduling, space, conference room, and scheduling script with videos for some of these items.

- Peer-reviewed articles that provides the trainees with the literature discussing the importance of HF-SMAs.

   <u>Adaptation:</u> Information on all aspects needed to execute a HF-SMA clinic was added to reduce trainee site startup time, including: four presentation templates, handouts from various societies concerning heart failure (30 items), a folder with scheduling information (7 items), spreadsheet for tracking metrics, HF education handbook, a folder of various administrative items (21 items), and peer-reviewed articles (17 articles).

   The following table summarizes the adaptations of the HF-SMA training program (Table 4).

### Discussion

In this formative evaluation, we noted the success of the virtual format and identified six themes that resulted in subsequent adaptations: 1) the two-day training was reduced to short online modules, 2) curriculum was adapted to fit local site's needs, 3) added periodic one-on-one coaching, 4) shifted training to focus on skills and knowledge needed for each team member requirements, 5) adapted curriculum provides for more team building during premeeting assignments, and 6) training had more information included.

   While providers faced the barrier of allocating time for training, but once in the training the trainees appreciated the experience of the trainers and found value in observing an actual SMA session. We identified program adaptations aligned with this theme of providers' need for access to relevant and tailored trainings that incorporated the expertise and areas of needed training for their clinical staff along with the needs of their patient population. Participants needed the training to be delivered efficiently and allowed participation flexibility. Training curriculum needed to include role-specific and Site-specific information that facilitated and

**Table 4. Curriculum adaptations.**

| Item | Original | Adapted |
|---|---|---|
| **Training time** | Two days (2) | Weekly (or determined by site's needs) |
| **Format** | Either in-person or virtual | Self-guided with virtual meetings |
| **Didactic** | ~ 4 hours | ~5 minute videos |
| **Lessons** | • Didactic completed in four hours<br>　○ Overview<br>　○ Benefits<br>　○ Getting Started<br>　　□ Leadership Support<br>　Provider Team<br>　Patient Population<br>　Goals & Guideline<br>　Space<br>　Charting/Billing Codes<br>　Session format<br>　Huddle<br>　Group Moderator<br>　Prescriber breakout<br>　Post Visit Summary<br>　○ Planning your timeline<br>　○ Manual & Resources Additional time for:<br>　　□ Site Needs Assessment (~2hrs)<br>　　□ View actual HF-SMA (~3hrs)<br>　　□ Mock SMA (~2hrs)<br>　　□ Question/Answer (~2hrs) | • Needs Assessment (completed by training team)<br>• Overview presentation (~ 30 mins.)<br>• Weekly 15–20 min meetings<br>• *Asynchronous Learning*<br>　○ Goals & Planning<br>　　□ Clinic flow, types*<br>　　□ Goals & Metrics*<br>　　□ Education*<br>　○ HF-SMA team members<br>　　□ Core Members*<br>　○ Administrative items<br>　　□ Create a Clinic*<br>　　□ Scheduling VVC*<br>　　□ Documenting the Visit*<br>　　□ Note Template<br>　○ HF-SMA patients<br>　　□ Patient Population*<br>　　□ Patient Recruitment*<br>　　□ HF Patient Report Viewer*<br>　○ Running a HF-SMA<br>　　□ Conducting an SMA*<br>　　□ Beginning a VVC appointment*<br>　　□ Facilitation*<br>　　□ Transitional HF-SMA*<br>　　□ HF-SMA practice (Mock)<br>* *denotes video presentations* |
| **Resources** | • Emailed PowerPoint (before training)<br>• Manual (after training) | • In private MS Teams group<br>• Available 24/7<br>　○ Videos<br>　○ Expanded detailed manual/guide<br>　○ Workbook<br>　○ Brochures<br>　○ Presentation Templates<br>　○ HF patient handouts<br>　　□ Informational<br>　　□ Medication lists<br>　○ Peer-reviewed articles<br>　○ Scheduling guides/information<br>　○ Various aids<br>　○ Various provider help guides<br>　○ Abbreviated & full videos |

supported team building. Providing needed and practical resources such as note templates, goal-setting worksheets, and suggestions for identifying program metrics helped sites get started.

## Virtual training to improve participation

Our study further supports an abundance of studies showing that virtual training education can be as successful as in-person training [21–26]. Virtual training allows for increased access and flexibility to participation. In turn, the adaptation to make the curriculum online to be viewed at the trainee's convenience should assist the productivity of any video mentoring meetings, reduce the provider burden, and increase patient satisfaction.

The challenge of the pandemic and ongoing challenges of access to ongoing professional development opportunities highlight the need for online virtual learning that includes both synchronous [27] and asynchronous learning formats. The virtual learning improves access, efficacy, cost effectiveness and ease of participation [28]. During COVID most continual education programs pivoted to a virtual format and revealing barriers and facilitators to this switch [29]. Barriers were technical barriers and the need for design adaptations to fit the virtual platform. The advantages of this format include convenience, reduction in travel time for rural participants, a suitable learning platform for participants, collaborative environment, and cost-effective [6–9]. The pandemic generated innovative ways to present information to students and improved students' evaluator of their trainers and students' motivation for learning [30].

## Learning in shorter sessions to improve effectiveness, flexibility, and efficiency

Similar to many other studies we found that presenting and consuming trainings in short time segments provides flexibility for trainees and may improve learning. Trainees can tailor the amount of information to fit their schedule and their learning preferences; these adaptations align with adult learning principles [31] that promote self-directed learning to supplement synchronous learning.

## Relevance: Needs assessment

Strategies to improve upon the traditional educational lecture encourage student participation that tailors the training to fit the needs and resources of the trainees and their setting allowing student to assess their existing resources to plan for and assess their own needs [32]. This attention to existing resources includes understanding each team member's knowledge of HF self-management, group facilitation skills, and experience working with their planned SMA clinical team. Trainees need to know required and optional individual team roles; this incorporates trainees to develop a program that is best suited for them by incorporating their teams' knowledge and skills. HF-SMA value of one-on-one coaching can be effective in changing the practices of healthcare professionals [33].

These results provide guidelines for designing virtual team-based training for a wide array of teams. Our data highlight the importance of understanding the needs and resources of the trainees. Virtual training programs are most effective when adapted to fit the trainees' needs and resources. Content should be presented in a format that fits the trainees' schedule and meets the needs of both the learner and the setting. By including practical resources sites felt supported and were motivated to get started. We shared the adapted training program as a resource for others interested in starting their own SMAs and as a methodological guide to improve implementation of other training programs.

## Limitations

This study has a few limitations, first was the effects of COVID-19 pandemic, as we were unable to hold the in-person training for Site-1 that would have enabled a comparison of the two training techniques. Nevertheless, the rich data from the virtual training interviews of Site-2 and observations, enabled the training group to adapt the training to better fit into the responsibilities of HF providers for future sites.

An additional limitation was the inability to secure a second site for an in-person comparison. The adapted curriculum placed the site assessment prior to training, as the training team had determined that if the site did not have HF Clinic experience, the implementation of a HF-SMA would be more difficult and frustrating for the site. However, this limitation led to

an adaptation that increased the effectiveness of the training and reduced the time from creating to executing a HF-SMA. Through utilizing this change, three sites were provided direction on what was necessary for a complete foundation prior to this training. The fifth site began the training less than two weeks prior to the project ending, therefore there was limited feedback on the revised curriculum.

Only one site was able to be fully trained on each version of the curriculum, which limits the generalizability of the results across training modalities or professional backgrounds (e.g., MD, NP, PharmD, etc.), to determine if there were varying perspectives on the training's efficacy. A larger study should recruit a variety of sites and providers to generate and support sustainable HF-SMA teams.

## Conclusions

We have taken the first step in identifying an improved model for a HF-SMA training program. A larger study using comparative designs to measure success and sustainability are needed.

## Supporting information

**S1 Appendix. Pre-training interview guide.**
(PDF)

**S2 Appendix. Post-training interview guide.**
(PDF)

**S3 Appendix. Post-implementation interview guide.**
(PDF)

## Acknowledgments

The study team would like to acknowledge David Aron, MD, MS (retired) for his design and development of the diabetes SMA, developing the qualitative team's skills, and sharing the knowledge with Dr. Wen-Chih Wu. Additionally, we would like to acknowledge the contributions of Melanie Parent and Troo Tucker for their efforts within this project.

## Disclaimer

The contents of this publication does not represent the views of the U.S. Department of Veterans Affairs or the United States Government.

## Author Contributions

**Conceptualization:** Sherry L. Ball, Julie Gee, Sharon LaForest, Kimberley Schaub, Tracey Taveira, Wen-Chih Wu.

**Data curation:** Rene Hearns.

**Formal analysis:** Rene Hearns, Sherry L. Ball.

**Funding acquisition:** Sherry L. Ball, Wen-Chih Wu.

**Investigation:** Sherry L. Ball, Tai-Lyn Wilkerson, Wen-Chih Wu.

**Methodology:** Sherry L. Ball, Julie Gee, Sharon LaForest, Kimberley Schaub, Tracey Taveira, Wen-Chih Wu.

**Project administration:** Rene Hearns.

**Supervision:** Sherry L. Ball, Tai-Lyn Wilkerson, Wen-Chih Wu.

**Validation:** Sherry L. Ball.

**Writing – original draft:** Rene Hearns, Sherry L. Ball.

**Writing – review & editing:** Rene Hearns, Sherry L. Ball, Tai-Lyn Wilkerson, Julie Gee, Sharon LaForest, Kimberley Schaub, Tracey Taveira, Wen-Chih Wu.

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
