## [Decision Letter · Decision Letter 0]

4 Sep 2024

Training providers to implement heart failure shared medical appointments: A qualitative evaluation

PONE-D-24-07219

Dear Dr. Hearns,

We’re pleased to inform you that your manuscript has been judged scientifically suitable for publication and will be formally accepted for publication once it meets all outstanding technical requirements.

Kind regards,

Richa Gupta

Academic Editor

PLOS ONE

EDITORIAL DECISION

This manuscript highlights the need for larger studies regarding the issue of heart failure self medication and is quite interesting. Based on reviewer’s comments following decision has been made: ACCEPT

Please find attached reviewer’s comments below:

REVIEWER 1: ACCEPT

Very well presented manuscript regarding heart failure self medication. Heart failure is a chronic disease and self care at home is very important for syntpom management and improvement. This manuscript highlights the need for larger studies regarding this issue and is quite interesting. Manuscript may be published with no revisions from my side. Rather interesting amd well presented Highlighting the need of establishment of standarized protocols and assistance for self care of patients which is crucial for better symptom control.

REVIEWER 2 - ACCEPT

Thank you for the opportunity to comment on this paper. The topic of this paper is interesting and there is a need for strategies to improve heart failure education efficiently with the growing population of chronic heart failure.

Summary: This paper touches on an important topic in the chronic care of the growing heart failure patient population where there is a deficit of providers and a need to deliver care more efficiently through shared medical appointments. This study administered by the VA aims to evaluate how providers can be trained to deliver SMAs. The paper discusses the initial strategy the research team had for delivery and the adaptations they made to their training model to improve their model as they encountered challenges. Overall, this is a paper offers a reasonable training model to disseminate SMA models in the heart failure population.

I have some specific and minor suggestions.

Introduction: Well written and provides robust background into the topic of SMA and the scope of the problem.

Line 81 “exist within the VHA..” there are two periods.

Methods: The methodology is sound and clearly explained. Pg 10 line 188 has two commas

Results:

The results have an interesting format with quotations of responses from various parties involved in the study. It is well written.

Discussion:

This is also well written.

Reviewers' comments:

Reviewer's Responses to Questions

**Comments to the Author**

1. Is the manuscript technically sound, and do the data support the conclusions?

Reviewer #1: Yes

Reviewer #2: Yes

2. Has the statistical analysis been performed appropriately and rigorously? 

Reviewer #1: Yes

Reviewer #2: N/A

3. Have the authors made all data underlying the findings in their manuscript fully available?

Reviewer #1: Yes

Reviewer #2: Yes

4. Is the manuscript presented in an intelligible fashion and written in standard English?

Reviewer #1: Yes

Reviewer #2: Yes

5. Review Comments to the Author

Reviewer #1: Very well presented manuscript regarding heart failure self medication. Heart failure is a chronic

Disease and self care at home is very important for syntpom management and improvement.

This manuscript highlights the need for larger studies regarding this issue and is quite interesting.

Reviewer #2: Thank you for the opportunity to comment on this paper. The topic of this paper is interesting and there is a need for strategies to improve heart failure education efficiently with the growing population of chronic heart failure.

Summary: This paper touches on an important topic in the chronic care of the growing heart failure patient population where there is a deficit of providers and a need to deliver care more efficiently through shared medical appointments. This study administered by the VA aims to evaluate how providers can be trained to deliver SMAs. The paper discusses the initial strategy the research team had for delivery and the adaptations they made to their training model to improve their model as they encountered challenges. Overall, this is a paper offers a reasonable training model to disseminate SMA models in the heart failure population.

I have some specific and minor suggestions.

Introduction: Well written and provides robust background into the topic of SMA and the scope of the problem.

Line 81 “exist within the VHA..” there are two periods.

Methods:

The methodology is sound and clearly explained. Pg 10 line 188 has two commas.

Results:

The results have an interesting format with quotations of responses from various parties involved in the study. It is well written.

Discussion:

This is also well written.

6. PLOS authors have the option to publish the peer review history of their article (what does this mean?). If published, this will include your full peer review and any attached files.

Reviewer #1: **Yes: **Dimitrios Afendoulis

Reviewer #2: No

---

## [Editor Report · Acceptance letter]

20 Sep 2024

PONE-D-24-07219 

PLOS ONE

Dear Dr. Hearns, 

I'm pleased to inform you that your manuscript has been deemed suitable for publication in PLOS ONE. Congratulations! Your manuscript is now being handed over to our production team.

Kind regards, 

on behalf of

Dr. Richa Gupta 

Academic Editor

PLOS ONE